# Vascular Plants from the Journey through Portugal (1797–1801) by Hoffmannsegg and Link at the Herbarium of the Real Jardín Botánico of Madrid

**DOI:** 10.3390/plants11182438

**Published:** 2022-09-19

**Authors:** Leopoldo Medina, Carlos Aedo

**Affiliations:** Real Jardín Botánico, CSIC, Plaza de Murillo 2, 28014 Madrid, Spain

**Keywords:** historical herbarium collections, original materials, typification, history of botany, hoffmannsegg, link, portugal

## Abstract

During the journey through Portugal by Hoffmannsegg and Link (1797–1801), these authors collected an appreciable number of specimens, most of which have been lost. Their collections are relevant since they were used by themselves or by other authors to describe numerous species. In the herbarium of the Real Jardín Botánico of Madrid, 70 specimens from this journey have been located. In the archive of this institution the letters that Hoffmannsegg and Link sent to Cavanilles accompanying these plants have also been located. The analysis of these letters, the herbarium labels and of the protologues has permitted to establish that 15 specimens are original material, four of which had already been proposed as lectotypes by other authors (*Airochloa caudata* Link, *Silene fuscata* Link ex Brot., *Silene micrantha* Link ex Otth and *Silene pernoctans* Link). The designation of a neotype for *Stipa gigantea* Link should be superseded, because an original material has been found. Thus, a lectotype for this taxon is proposed.

## 1. Introduction

The naturalist and botanist Johann Centurius Graf von Hoffmannsegg (Dresden, 1766-Dresden, 1849) [1] promoted two scientific expeditions to Portugal at his expense. The first one took place between the fall of 1795 and the spring of 1796, in the company of the German naturalist Wilhelm Gottfried Tilesius. This first expedition had little success, according to Gomes Oliveira [2], due to the difficulties they encountered in obtaining permission from the local authorities to travel through the country. On this occasion they barely toured the environs of Lisbon.

On the second trip, he was accompanied by Johann Heinrich Friederich Link (Hildesheim, 1767-Berlin, 1851) [3]. This journey began in Hamburg on 18 August 1797, heading for Lisbon, but due to poor sea and ship conditions, the sea voyage stopped at Dover, from where the expeditionaries went to Calais. They crossed France and Spain to enter Portugal through Elvas on 11 February 1798. During their trip they met Antonio José Cavanilles (Valencia, 1745-Madrid, 1804) in Madrid, from whom they received the gift of a “*… grand nombre de plantes choisies de l’Espagne*” [4] (p. 2). Hoffmannsegg and Link initially went to Lisbon where they obtain a generous safe-conduct that allowed them and their servants to travel freely throughout Portugal [2].

From this base camp they carried out movements and collections throughout the country, from the Algarve, in the extreme south, to Gerês in the vicinity of the Spanish border with Galicia. In Coimbra they meet Brotero with whom they later exchanged plants. They worked in Cabo da Roca, Alentejo, Setúbal, Serra da Arrábida, Coimbra, Aveiro, Porto, Serra da Estrela, etc. until the spring of 1799, at which point Link had to return because the license given to him by the University of Rostock was expiring. He did so via England, where he compared the Portuguese plants with those in J. Banks’s herbarium. Hoffmannsegg continued to tour the country, from Bragança to Alentejo, in part reinforcing his work in localities already visited as well as in others where they had not yet collected, until August 1801, when he returned to Germany by sea.

Hoffmannsegg and Link [4] (p. 8) advance, in the preface to their Portuguese flora, their intention to deposit their herbarium in a public institution to be determined: “*Lorsque l’ouvrage sera terminé, les curieux pourront inspecter tout l’Herbier de plantes Portugaises qui en a fait la base principale, soit chez le COMTE DE HOFFMANSEGG, soit dans quelqu’institut public qu’on aura soin d’indiquer*”. Unfortunately, we still do not have reliable and complete data on the history of the herbariums of these authors, especially with regard to Hoffmannsegg’s herbarium. The Willdenow plants collection was bought in 1818 for the Berlin herbarium, at the behest of Friedrich Wilhelm III, King of Prussia [5]. According to Urban [6], Willdenow had acquired 470 plants from the Hoffmannsegg herbarium which would now be preserved in B-W. Most of these materials were collected by Hoffmannsegg’s servant Friedrich Wilhelm Sieber in Brazil, and therefore only a small part corresponds to the Portuguese expedition. As Robert Vogt [7] kindly indicates, the database of this herbarium is still in the process of being revised as far as the names of the collectors are concerned, so the number of Portuguese specimens of this collector cannot be accurately determined. We have identified in the Berlin herbarium (B), through JSTOR Global Plants [8] and the Berlin Virtual Herbarium website [9], five specimens that correspond to Hoffmannsegg collections with the sole indication “Habitat in Lusitania”.

D.F.L. Schlechtendahl, who was the director of the Botanical Gardens in Halle between 1833 and 1886, obtained a large number of duplicates from B which he transferred to the Halle herbarium (HAL) [10,11]. That is probably the origin of the Hoffmannsegg specimens from Portugal of HAL, of which there are currently 30 accessible through the JACQ online herbaria database [12], and 27 of them on the JSTOR Global Plants [8] website. The Geneva herbarium (G and G-DC) [13] houses 21 specimens attributed to Hoffmannsegg from which only four clearly correspond to the Portuguese expedition, three of them belong to the de Candolle Herbarium (G-DC), and one is included in the General Collection (G). Through the JSTOR Global Plants [8] website, four Portuguese specimens have also been located, all of the genus *Thymus*, attributed to Hoffmannsegg in the Hamburg herbarium (HBG), and another one in Montpellier herbarium (MPU). Other non-European materials are kept in DR, P, PH and W. Stafleu & Cowan [14] (p. 245) indicate that there are also plants from Hoffmannsegg in the Helsinki herbarium (H), which has not been confirmed through the Natural History herbarium (H) database [15]. Although the available data are incomplete, everything points to the fact that very few of the specimens he collected in Portugal have survived to the present day.

Link was appointed director of the Botanischer Garten und Botanisches Museum Berlin in 1815, a date assumed to be the founding of the Berlin institutional herbarium [5,6]. Link’s personal herbarium was acquired by the Botanischer Garten und Botanisches Museum Berlin in 1851, but was mostly destroyed during Second World War, although, for example, the Pteridophyta were preserved [5,16]. In the database of the Berlin herbarium (B) [9] we have located five specimens attributed to Link. They are also included on the JSTOR Global Plants website [8] and clearly correspond to Portuguese localities. The online database of the Geneva herbarium (G and G-DC) [13] shows two specimens from this author’s expedition to Portugal, also accessible through JSTOR Global Plants website [8], one of which is labeled “Herb Pavon”, so it was probably part of the herbarium that Pavón’s heirs sold to Boissier [17].

The historical section of the University of Seville herbarium (SEV) keeps two Lusitanian plants attributed to Link [18], undoubtedly coming from the herbarium that Claudio Boutelou formed when he was in the Real Jardín Botánico of Madrid [19]. Among all the herbaria included in the JACQ online database [12], we could only find one specimen attributed to Link in Halle herbarium (HAL). We have also located two specimens of Link’s Portuguese grasses in Komarov Botanical Institute herbarium (LE) through the JSTOR Global Plants website [8]. The Natural History Museum herbarium (BM) preserves four type specimens of Link plants, three of them accessible through JSTOR Global Plants [8] and the other one on its Data Portal website [20]. The JSTOR Global Plants website [8] offers information about a *Crocus clusii* J. Gay specimen housed in the Kew herbarium (K) that can clearly be attributed to this author, to add to another *Verbascum* specimen found in The Linnaean Society herbarium (LINN). The Herbarium Hamburgense (HBG) website [21] and the Finnish Museum of Natural History herbarium (H) database [15] give information with regard to a different specimen collected by Link. Stafleu & Cowan [16] (p. 655) indicate that there are also Link specimens in BR, C, FI, LIV, P, PH and W, which has not been confirmed, at least regarding Portuguese plants.

The numbers of plants of these authors obtained from the literature and from the available databases must be taken with some caution since the digitization process of the herbarium collections is still in progress in many of this herbaria. Link [22] (p. 238) notes that “*Nous avons trouvé en Portugal 1532 espèces de plantes ordinaires, 572 espèces de plantes cryptogamiques*”, which indicates that they collected a higher number of specimens and seems to suggest that most of the collections that these authors gained in Portugal have been lost.

Hoffmannsegg and Link initially published articles in German journals to publicize their findings in Portugal as a previous step to their main work. Among the most relevant, since they contain taxonomic novelties, we could mention Link [23,24,25,26,27,28] and Hoffmannsegg and Link [29]. Brotero in 1804 published his *Flora Lusitanica*, where he recognized the plants communicated by Hoffmannsegg and Link to him, which apparently was not reciprocal [30,31].

The main work of Hoffmannsegg and Link, the *Flore Portugaise*, was published in two volumes with independent pagination. The first [4], to which fascicles 1 to 14 correspond, was published between 1 September 1809 and 1820. The second volume [32], to which fascicles 15–22 correspond, was published between 1813 and 1840 [14] (p. 246). The work was left unfinished. According to Gomes Oliveira [2] (p. 114), the editorial plan included a third volume with eight issues. This flora has detailed descriptions in French and Latin, includes 109 splendid plates and mentions precise localities of most of the species. Gomes Oliveira [2] (p. 17) points out that 659 species are described, that is, about a quarter of those known today in mainland Portugal [33] (p. 112). The *Flora Lusitanica* of Brotero [34] describes 1225 species [2] (p. 136), which is slightly less than half of those known today [33] (p. 112).

Parallel to the *Flore Portugaise*, Link [35,36,37,38] continued to publish taxonomic novelties based on the Portuguese collections. Other authors then had access to the plants collected by Hoffmannsegg and Link and also published new species based on them [39,40,41].

An indication of the importance of the work of Hoffmannsegg and Link is its contribution to the description of Iberian endemisms. According to Buira and Aedo [42], they are among the twelve most prolific authors in terms of the description of endemic species. In addition, they are, together with Brotero, the main responsible for the description of the endemisms of the western Iberian Peninsula.

The objective of this work is to make known the Portuguese plants of Hoffmannsegg and Link that are preserved in the Herbarium MA of the Real Jardín Botánico of Madrid, and to establish if they constitute original material of the names proposed by the aforementioned authors.

## 2. Materials and Methods

The archive of the Real Jardín Botánico of Madrid keeps six letters sent by Hoffmannsegg to Cavanilles from Lisbon between October 1798 and August 1801 [43,44,45,46,47,48], and two letters sent by Link also to Cavanilles from Lisbon, in October 1798 [49] and in January 1799 [50].

On the other hand, a review of the nomenclatural database of the *Flora iberica* project has been made to detect the names of the taxa described by Hoffmannsegg and Link. Only those heterotypic names validly published have been considered and, in the case of illegitimate names, those that are by homonymy.

From these two sources of information, the Portuguese plants that Hoffmannsegg and Link sent to Cavanilles have been located in the MA herbarium, with the additional help of the herbarium database. The currently accepted name is indicated if it is different from the original. In the corresponding section, it is discussed whether there is sufficient evidence to consider that these Hoffmannsegg and Link plants are original material of a validly published name. Proposing lectotypes has been avoided, except in exceptional cases of groups well known to the authors, in accordance with Recommendation 9A.2 from the International Code of Nomenclature [51] (p. 26): “Designation of a lectotype should be undertaken only in the light of an understanding of the group concerned”, to avoid possible future confusion and further changes. It is necessary to keep in mind that original materials corresponding to the names of Hoffmannsegg and Link may still appear in various European herbaria. It is mentioned whether other authors have proposed any lectotype on these plants.

We have adopted the volume and fascicle numeration as well as the dates proposed by IPNI to properly and unequivocally cite the *Journal für die Botanik*.

## 3. Results and Discussion

### 3.1. Hoffmannsegg and Link’s Letters to Cavanilles

Hoffmannsegg’s first letter to Cavanilles is dated 14 October 1798 [43]. In it, he announces that he and Link are preparing to send a first set of 50 plants so that Cavanilles can give them his opinion about their identity. In this letter he establishes the working method through a numbered list of the specimens that would be included together with the shipment, with a copy in the hands of the German botanists: *“… Comme j’ai gardée une copie de la liste- Ci-jointe, Vous n’avez qu’a nous rapporter seulement au no. de cells sur lesquelles il Vous plaira de nous instruire*” (Figure 1). He also announces a second similar shipment coming four to six weeks later. The attached list is not included in this letter. However, in Link’s letter to Cavanilles, dated 12 October 1798 [49], despite its earlier facial date, it refers to the aforementioned Hoffmannsegg letter and includes the announced list of 50 plants. Cavanilles notes his opinions in the margin of the letter received. For example, he adds: “*Sit; venumtamen flores sunt avena elatioris, a qua foliis difert*” to “*35. Avena pallens nob.*”, “*Verbenacam credo*” to “*43. Salvia clandestine*” or “*bene*” to “*44. Pinguicula lusitanica*”, to correct or accept the identifications of the German botanists. It goes without saying that Cavanilles would reply to the letter with this information.

In the following letter from Link, dated 28 January 1799 [50], a list with plants numbered from 51 to 102 is attached, to which Cavanilles adds his comments in the same way. In the remaining letters from Hoffmannsegg, dated 14 April 1799 [45], 1 February 1801 [46], 6 July 1801 [47], and 16 August 1801 [48], he is thankful for the determinations and diverse news that are given, but without lists of plants. The only other list available is the one received on 27 April 1801 [44], with no attached letter available in which another shipment of plants numbered from 1 to 45 is recorded.

This correspondence mentioned above is obviously incomplete, but sufficient to understand the working method agreed upon by these authors. In all probability, an indeterminate part of the letters from German botanists to Cavanilles has not reached the present day, since part of the plants sent do not appear in any of the three available lists. It is interesting to indicate that the documentation on Cavanilles remained in the hands of his heirs for almost 200 years until 1992, when it was deposited in the archive of the Real Jardín Botánico of Madrid [52].

### 3.2. The Portuguese Plants of Hoffmannsegg & Link at the Herbarium MA

The current sheets where the Hoffmannsegg and Link plants have been mounted have small labels of approximately 5–10 × 3–5 cm. A comparison between the data from the letters and those from the numbered labels supports the idea that the sheets were sent with only a number annotated by the German botanists and that Cavanilles added his comments in clearly distinguishable handwriting and ink. This portion of the original folded paper sheet was cut out at some point and used as a label. A small part of the sheets that are not mentioned in the letters seem to have been totally or partially relabeled by a hand other than Cavanilles (Figure 2).

Labels usually lack a precise locality, since the lists sent by Hoffmannsegg and Link generally do not include this data. Cavanilles adds “*ex Lusitania*” or an equivalent text, plus the name of the plant. Exceptionally in MA00114705, Cavanilles writes “*Near the Tagus in Portugal*”, translating the information from the letter [49] in which Link says “*14. Lathraea phelypaea In arenosis/trans Tagum*”. Other times Cavanilles does not transpose the locality (*22. Rhinanthus versicolor Lamark. Frequens circa Olyssiponem*) from Link’s letter [49] to the label, and simply notes “*Lusitania*”.

The collection’s 70 vascular plants of Hoffmannsegg and Link located on the Herbarium MA are listed below (Table 1), with the MA herbarium barcode number and an indication of the current accepted name if it differs from the original.

### 3.3. Original Materials of the Taxa Described by Hoffmannsegg and Link in the Herbarium MA

Below, in alphabetical order, the original materials located in MA of the taxa described on specimens collected by the Hoffmannsegg and Link expedition to Portugal are listed. Its connection with the letters sent to Cavanilles is indicated, if it exists, and its nomenclatural and taxonomic status is discussed.

*Airochloa caudata* Link, Linnaea 17: 405 (1843). [*Koeleria crassipes* Lange, Pugillus. Pl. Hispan. 1: 43 (1860)]. Link [38], when describing the species, indicates its origin: “*Plantam olim in Lusitania legimus, supra Fundão, in castanetis humidis*”. In the letter sent by Hoffmannsegg to Cavanilles dated 27 April 1801 [44], it reads “No. 20. Aira/si n. sp. Caudata”, but no locality is indicated. On the label of MA00177284, which bears the number “No. 20.” handwritten by Hoffmannsegg, Cavanilles notes “Aira caudata Hoffmansegg/an vere aira?/ex Lusitania”. Quintanar & Castroviejo [53] (p. 1053) designate this specimen as a lectotype. Dates of publication of the Linnaea are according to Foster [54].*Genista bracteolata* Link, Enum. Hort. Berol. Alt. 2: 224 (1822) [*G. micrantha* Gómez Ortega, Nov. Pl. Descr. Dec.: 68, Table 10 Figure 1 (1798)]. Link [35] publishes this species without indicating its origin. Sprengel [55] (p. 176) picks up the name in the synonymy of *Spartium stylosum* Spreng., for which he indicates “Lusitan”. In the letter sent by Hoffmannsegg to Cavanilles dated 27 April 1801 [44], it reads “No. 40. Genista bracteolata. Link” but no location is indicated. On the label of MA00059207, which bears the number “No. 40.” handwritten by Hoffmannsegg, Cavanilles notes “Spartium/Genista bracteolata Link e Lusitania”. This sheet is original material.*Isatis platyloba* Link ex Steud., Nomencl.: 440 (1821). Steudel [56] gives a new name for *Isatis lusitanica* Brot., Fl. Lusit. 1: 560 (1804), since this epithet had previously been used by Linnaeus: “I. platyloba. Link. I. lusitanica. Brot (not L.)”. However, Brotero (1804) attributes the plant to Hoffmansegg: “Comm. primum á Cl. Com. Hoffmannsegg” and does not mention Link. The species had been discovered by Hoffmannsegg in May 1800, in Miranda do Douro (the only locality mentioned by Brotero), when Link had already returned to Germany [2] (p. 344). In fact, Link [22] (p. 38), when recounting the part of the trip to Portugal in which only Hoffmannsegg participated, says: “*Les rochers qui bordent le Douro sont intéressans pour la botanique; nous y vîmes fleurir une nouvelle espèce d’Isatis*…” In the letter sent by Hoffmannsegg to Cavanilles dated 27 April 1801 [44], it reads “No. 45. Isatis”, but no locality is indicated. On the MA00045311 label bearing the writing “No. 45.” and handwritten by Hoffmannsegg, Cavanilles notes “Isatis armena Linn/ex Lusitania misit Link”. This sheet is original material (Figure 3). On the other hand, no plant of the *Isatis* genus is preserved in the herbaria of Brotero and Valorado [57]. Considering what Brotero [34] said, it seems more logical to consider Link’s mention as an error. Consquently, the name of the species should be corrected as *Isatis platyloba* Hoffmanns. ex Steud., Nomencl.: 440 (1821).*Linaria polygalifolia* Hoffmans. & Link, Fl. Portug. 1: 248, pl. 44 (1811). The authors describe the species from “*Dans les contrées maritimes, prés de Coimbre, O-Porto, S. Martinho* [pr. Setubal, 38°32’N, 08°30’W]”. In the letter sent by Link to Cavanilles in October 1798 [49], it reads “9. Antirrhinum Paralias nob. … In arenosis maritimis prope Setuval” handwritten by Link, in whose margin an indication of Cavanilles is added: “S. nova mihi unknown/Videtur ant. Lusitanicum/Lamarck dict. t. 4. p 361”. The label of MA001098441 does not bear the number that appears on the letter, but Cavanilles notes “Linaria lusitanica/ant. lusitanicum Lamarck/Dict. Vol. 4. p. 361/ex Lusitania. Cavanilles dedit”, which is the same note that he himself added in the right margin of the letter. This sheet is original material.*Orchis intacta* Link, J. Bot. (Schrader) 2(2): 322 (Apr. 1800) [*Neotinea maculata* (Desf.) Stearn in Ann. Mus. Goulandris 2: 79 (1975)]. Link [23] describes the species without indicating locality. In the letter sent by Link to Cavanilles in January 1799 [50], it reads “N. 51. Orchis intacta nova ut vidit species”, but no locality is indicated. On the label of MA00051515, which bears the number “51” handwritten by Link, Cavanilles notes “Orchis intacta Link/ex Lusitania”. This sheet is original material.*Orchis longicruris* Link, J. Bot. (Schrader) 2(2): 323 (Apr. 1800) [*O. italica* Poir. in Lam., Encycl. 4: 600 (1798)]. Link [23], when describing the species, indicates that it has “*Labium trifidum, lacinia media trifida, cuius laciniae lateral longissimae, angustissimae, intermedia brevior*”, as corresponds to *O. italica* and indicates as locality “*Häufig auf den Hügeln um Bellas*”. This geographical reference does not appear in the available letters. The label of the MA00023704 sheet reads “ex Lusitania/Link dedit”. In the Berlin herbarium (B-W-16824-01 0) there is a specimen attributed to Link, from Portugal, (“Habitat in Lusitaniae collibus”) that would also turn out to be original material.*Quercus australis* Link ex Spreng., Handbuch 2: 466 (1831) [*Quercus faginea* subsp. *broteroi* (Cout.) A. Camus, Chenés 2: 179 (1939). Link [36] describes the species “In Lusit.”, without specifying the locality. In the letter sent by Link to Cavanilles in January 1799 [50] it reads “N. 86. Quercus australis”, but no locality is indicated. On the label of MA00174257, which bears the number “86” handwritten by Link, Cavanilles notes “Quercus australis/e Lusitania. Link”. This sheet is original material.*Sedum pruinatum* Link ex Brot., Fl. Lusit. 2: 209 (1804). Brotero [34] describes the species as “*Hab. in Gerez, et ad Rio Homem in Duriminia…*
*Planta viva nondum mihi occurrit: ex specimine sicco a Cl.*
*Prof. Link communicato*”. In the letter sent by Link to Cavanilles in January 1799 [50] it reads “N. 65. Sedum pruinatum nob.”, but no locality is indicated. On the label of MA00051515, which bears the number “65” handwritten by Link, Cavanilles notes “Sedum pruinatum/Link e Lusitania”. On the other hand, no plant corresponding to this species is preserved in the herbaria of Brotero and Valorado [57]. This sheet is original material.*Silene fuscata* Link ex Brot., Fl. Lusit. 2: 187 (1804). Brotero [34] describes the species as “*Hab. in collibus circa Cabeça de Montachique, necnon circa Obidos, et al.ibi in Extremadura… primum a Cl.*
*Prof. Link*”. In the letter sent by Link to Cavanilles (AJB, Div. XIII, 3, 40, 2) in January 1799, it reads “N. 68. Silene fuscata nob.”, but no locality is indicated. On the label of MA00031723, which bears the number “68” handwritten by Link, Cavanilles notes “Silene fuscata Link/e Lusitania”. Talavera and Muñoz Garmendia [58] (p. 435) designated this sheet as a lectotype. In the JSTOR database, a specimen collected in Algeria by W. Schimper in 1832 that is kept in the Hamburg herbarium (HBG-503531) also appears as a type of *Silene fuscata*, which cannot be considered in any way as original material of this species. It would be interesting to see if the sheet mentioned by Coutinho [58] (p. 374) from the Valorado herbarium in LISU, where some plants sent by Brotero are preserved, can also be considered original material.*Silene micrantha* Link ex Otth in DC., Prodr. 1: 372 (1824). [≡*Silene micropetala* Lag., Varied. Ci. 2(4): 213 (1805)]. A. Otth describes in Candolle [59] the species as “Link Cav. herb./in Lusitania”. In the letter sent by Link to Cavanilles in January 1799 [50] it reads “N. 70. Silene micrantha nob.”, but no locality is indicated. On the label of MA00031016, which bears the number “70” handwritten by Link, Cavanilles notes “Silene micrantha Link/e Lusitania”. When Lagasca [60] describes *S. micropetala* he does not indicate any locality nor does he mention the epithet *micrantha*. Later, in Lagasca [61] (p. 15) when he refers to his *S. micropetala* again, he specifically mentions the plant that Link sent to Cavanilles with his locality plus a locality of his own: “*Silene micrantha. Link. Cav. Herb./Legi in tractu dicto Barranco hondo juxta viam, quae ducit ad tractum Salcedal non procul a Fuencarral oppido. Hab. in Lusitania. herb. Cavan*”. Talavera & Muñoz Garmendia [58] (p. 419) choose to typify both species on the material sent by Link, which does not make the name *S. micrantha* illegitimate according to the Article 52.2 note 2 of the International Code of Nomenclature [51] (p. 128), as such authors point out.*Silene pernoctans* Link in Spreng., Syst. Veg. ed. 16, 2: 408 (1825). [*Silene colorata* Poir., Voy. Barbarie 2: 163 (1789)]. Sprengel [40] attributes the species to Link and gives the origin “Lusitania?”. In the letter sent by Link to Cavanilles in January 1799 [50] it reads “N. 72. Silene perroctans n.”, but no locality is indicated. On the label of MA00031016, which bears the number “72” handwritten by Link, Cavanilles notes “Silene pernoctans Link/e Lusitania”. Talavera and Muñoz Garmendia [58] (p. 423) designated this sheet as a lectotype. In the Halle herbarium (HAL), a specimen attributed to Hoffmannsegg that possibly comes from the same collection (HAL-0012735), is preserved as a type of the species.*Statice pinifolia* Link ex Brot., Fl. Lusit. 1: 486 (1804) [*Armeria pinifolia* (Link ex Brot.) Hoffmanns. & Link, Fl. Portug. 1: 437 (1813–1820). Brotero [34] bases the description of his species on plants communicated by Link: “*Hab. ad ripas Sadão prope Setubal. Fl. aest. Per. Comm. a Cl. P. Link*”. In the letter sent by Link to Cavanilles in October 1798 [49] reads “48. Statice quae videtur nova species a St. Armeria sat diverse. St. pinifolia nob. In arenosis maritimis”. On the label of MA00174261, which bears the number “48” handwritten by Link, Cavanilles annotates “e Lusitania. Link”. This sheet is original material. The sheet consists of two fragments, a rosette of leaves and a fragment of the scape, but lacks the capitulum, which advises that an epitype be sought if it were typified. It would be interesting to check if the sheet mentioned by Coutinho [57] (p. 363) from the Valorado herbarium in LISU, where some plants sent to it by Brotero are preserved, can also be considered original material.*Statice pungens* Link, J. Bot. (Schrad.) 3(1): 60 (June 1800) [*Armeria pungens* (Link) Hoffmanns. & Link, Fl. Portug. 1: 439 (1813–1820)]. Link [24] describes the species from “*In den sandigen Gegenden zwischen Cabo Espichel und Porto Brandano*”. In the letter sent by Link to Cavanilles in January 1799 [50] it reads “N. 94. Statice pungens N.”, but no locality is indicated. On the label of MA00174257, which bears the number “94” handwritten by Link, Cavanilles notes “Statice pungens Link/ex Lusitania ab ipso misa”. This sheet is original material.*Stauracanthus aphyllus* Link, Neues J. Bot. 2(2): 52 (1808) [*Stauracanthus genistoides* (Brot.) Samp., Ann. Sci. Acad. Polytechn. Porto 7: 53 (1912)]. Link [28] describes the species “*circa Coina, Moita, Aldea galliga ut alibi trans Tagum prope Lisboa*”. In the letter sent by Link to Cavanilles in January 1799 [50] it reads “N. 73. Singularis Ulicis species, quam stauracanthum appellamus”, but no locality is indicated. On the label of MA00060058, which bears the number “73” handwritten by Link, Cavanilles notes “Ulex stauracanthus Link/e Lusitania misit autor”. This sheet is original material.*Stipa gigantea* Link, J. Bot. (Schrader) 2(2): 313 (Apr. 1800). Link [23] describes the species from “*An den Sandhügeln bei Setuval, Bellas, und am südlichen Abhange der Estrella fanden wir es nicht selten*”. This species does not appear in the available letters. On the label of the MA00005273 sheet, which bears the number “62” handwritten by Link, Cavanilles notes “Avena gigantea sp. n./ex Lusitania”. This specimen is connected to Link species through the epithet “*gigantea*” and fits very well with the current concept of *S. gigantea*. Vázquez & Barkworth [62] (p. 491) assumed that the original material had been destroyed in B and designated a neotype (HSS-9002). According to Article 9.19 of the International Code of Nomenclature [52] (p. 25): “…that choice should be superseded if (a) the holotype or… any of the original material is found to exist”. We designate here as lectotype the specimen MA00005273 (Figure 4), which contains a fragment with a well-developed inflorescence and allows us to unequivocally appreciate the distinctive characters of the species [63].

### 3.4. Types of the Taxa Described by Hoffmannsegg and Link in the Herbarium MA

As indicated above, the lectotypes of the names *Airochloa caudata* Link, *Silene fuscata* Link ex Brot., *Silene micrantha* Link ex Otth and *Silene pernoctans* Link were proposed by previous authors [53,58]. Vázquez & Barkworth [62] erroneusly proposed a neotype for *Stipa gigantea* Link. In this work, original material of *S. gigantea* has been located at MA, and consequently a lectotype has been proposed to replace the neotype.

## Figures and Tables

**Figure 1 plants-11-02438-f001:**
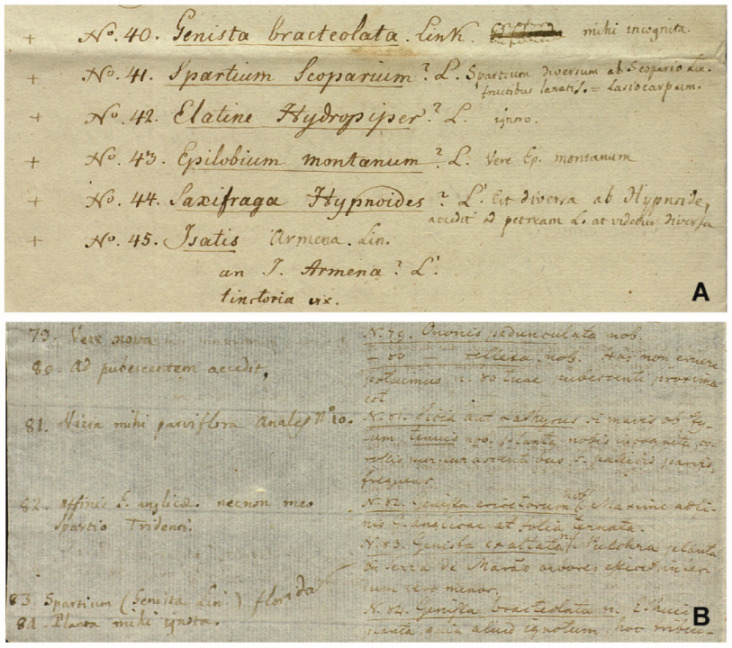
Letters to Cavanilles at the archive of the Real Jardín Botánico of Madrid. (**A**) Fragment of the letter (page 4) sent by Hoffmannsegg [44] in 27 April 1801; original list by Hoffmannsegg (left) with the later annotations of Cavanilles (right) and the check symbol (+) near the left edge. (**B**) Fragment of the letter (page 5) sent by Link [50] in 28 January 1799; original list by Link (right) and later annotations by Cavanilles (left) with the same number.

**Figure 2 plants-11-02438-f002:**
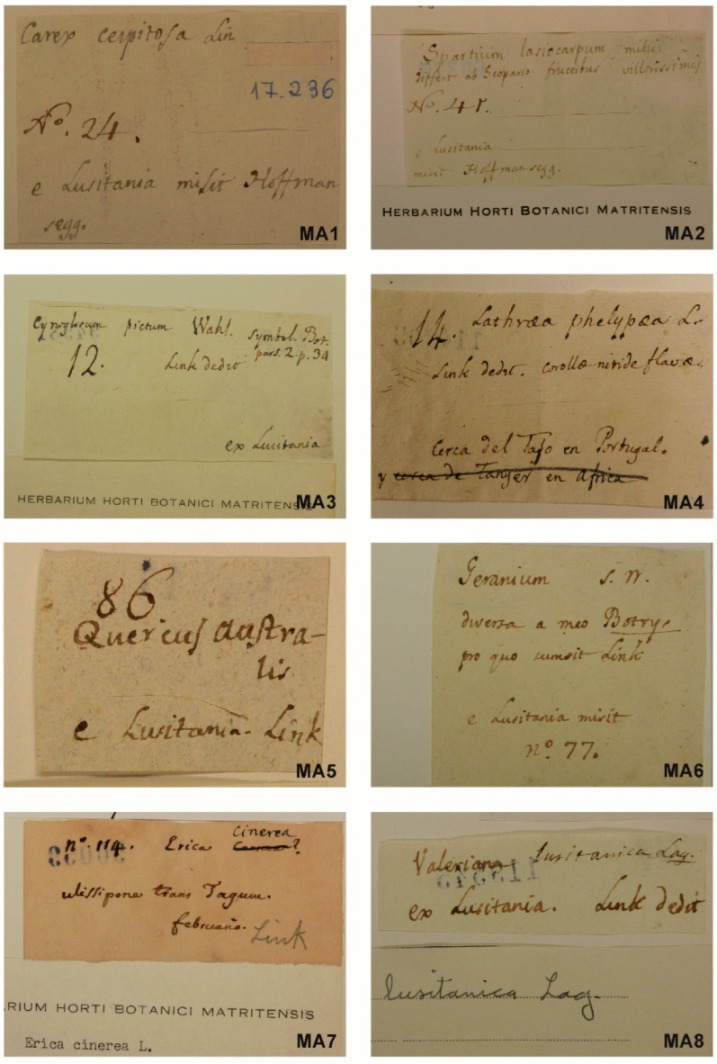
Examples of labels in the specimens of Hoffmannsegg and Link in the Herbarium MA: MA1 (MA017236) and MA2 (MA060375) from specimens sent by Hoffmannsegg. MA3 (MA094981), MA4 (MA094981), MA5 (MA114705) and MA6 (MA054711) from specimens sent by Link. MA7 (MA090059) and MA8 (MA119545) are from specimens sent by link but not in the letters [43,44,45,46,47,48,49,50] in the archive of the Real Jardín Botánico of Madrid.

**Figure 3 plants-11-02438-f003:**
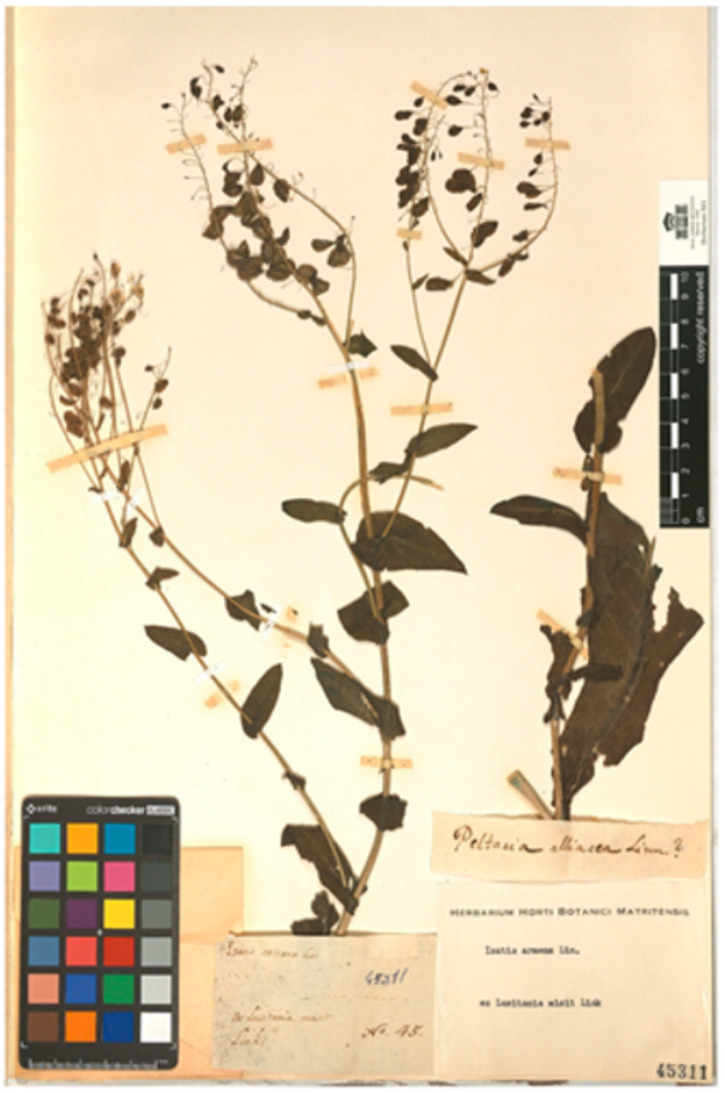
Specimen of *Isatis platyloba* Hoffmanns. ex Steud., sent by Hoffmannsegg to Cavanilles in 27 April 1801 (MA00045311).

**Figure 4 plants-11-02438-f004:**
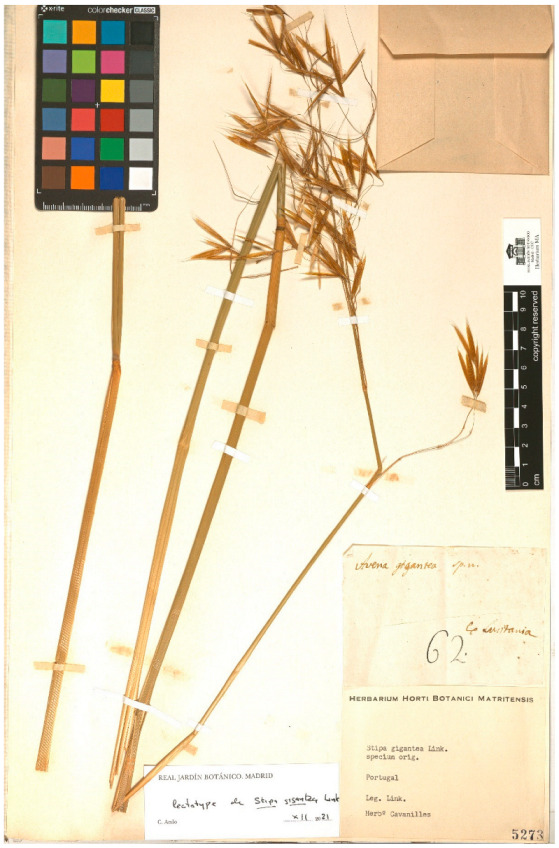
Lectotype of *Stipa gigantea* Link, in the Herbarium MA (MA00005273).

**Table 1 plants-11-02438-t001:** Plants in the Herbarium MA of the Real Jardín Botánico of Madrid, sent by J.C. Hoffmannsegg and J.H. Link to A.J. Cavanilles, from their expedition to Portugal (1797–1801). Specimens are sorted by its mention in each letter of reference, with those without a letter of reference at the end. Subsequently ordered by the ordinal in the original letter or in the label. Names or author corrections are between brackets. Original materials, current identification and the MA herbarium number is indicated.

Number in the Letter	Original Name in the Label	Original Material	Current Identification	MA Herbarium Number
**Letter from Hoffmannsegg, 27 April 1801 [44]**				
1	*Crambe corvini* Link		*Calepina irregularis* (Asso) Thell	MA045858
2	*Thlapsi praecox*		*Jonopsidium abulense* (Pau) Rothm.	MA045175
4	*Brassica cheranoides* Link		*Erysimum linifolium* (Pourr. ex Pers.) J. Gay	MA048975
6	*Narcissus herminicus* Hoffmansegg		*Narcissus rupicola* Dufour	MA148300
8	*Bunium*		*Conopodium pyrenaicum* (Loisel.) Miégev.	MA148935-1
9	*Bunium*		*Conopodium pyrenaicum* (Loisel.) Miégev.	MA148935-2
10	*Campanula primulaefolia* [*primulifolia*] Brotero		*Campanula primulifolia* Brot.	MA121775
14	*Prunella hastata* Link		*Prunella grandiflora* (L.) Scholler	MA101718
16	*Thymus acinos*		*Acinos rotundifolius* Pers.	MA174388
20	*Aira caudata* Hoffmansegg [*Airochloa caudata* Link]	Yes	*Koeleria crassipes* Lange	MA177284
24	*Carex cespitosa* [*caespitosa*] Lin.		*Carex elata* subsp. *reuteriana* (Boiss.) Luceño & Aedo	MA017236
35	*Euphorbia*		*Euphorbia oxyphylla* Boiss.	MA174147
39	*Genista exaltata*		*Genista florida* L.	MA059180
40	*Genista bracteolata* Link	Yes	*Genista micrantha* Ortega	MA059207
41	*Spartium lasiocarpum*		*Cytisus grandiflorus* (Brot.) DC.	MA060375
43	*Epilobium montanum* Lin.		*Epilobium montanum* L.	MA084026
45	*Isatis armena* Lin.	Yes	*Isatis platyloba* Link ex Steud.	MA045311
**Letter from Link, 12 October 1798 [49]**				
4	*Drosera lusitanica*		*Drosophyllum lusitanicum* (L.) Link	MA051110
9	*Linaria lusitanica*	Yes	*Linaria polygalifolia* Hoffmanns. & Link	MA109844
12	*Cynoglossum pictum* Wahl [Vahl]		*Cynoglossum creticum* Mill.	MA094981
14	*Lathrea phelypea* [*Lathraea phelypaea*]		*Cistanche phelypaea* (L.) Cout.	MA114705
15	*Cynoglossum cheirifolii* [*cheirifolium*]		*Cynoglossum clandestinum* Desf.	MA095056
16	*Anagalis monelli*		*Anagalis monelli* L.	MA091489
22	*Rhinanthus versicolor*		*Bartsia trixago* L.	MA114690
31	*Stipa paleacea* Vahl.		*Stipa capensis* Thunb.	MA005057
34	*Festuca cristata* L.		*Rostraria cistata* (L.) Tzvelev	MA009357
42	*Empetrum album*		*Corema album* (L.) D. Don	MA150430
44	*Pinguicula lusitanica*		*Pinguicula lusitanica* L.	MA149051
46	*Scirpus lasiophyllus*		*Scirpus michelianus* (L.) Link	MA016144
48	*Statice*	Yes	*Armeria pinifolia* (Brot.) Link	MA174261
**Link, 28 January 1799 [50]**				
51	*Orchis intacta* Link	Yes	*Neotinea maculata* (Desf.) Stearn	MA024413
55	*Cistus circunflexus*		*Helianthemum nummularium* (l.) Mill.	MA164340
65	*Sedum pruinatum*	Yes	*Sedum pruinatum* Brot.	MA051515
68	*Silene fuscata* Link	Yes	*Silene fuscata* Link ex Brot.	MA031723
70	*Silene micrantha* Link	Yes	*Silene micropetala* Lag.	MA031001
72	*Silene pernoctans* Link	Yes	*Silene colorata* Poir.	MA031016
73	*Ulex stauracanthus* Link	Yes	*Stauracanthus genistoides* (Brot.) Samp.	MA060058
74	*Dianthus attenuatus* ? Smith [Sm.]		*Dianthus* sp.	MA033833
75	*Geranium glandulosum* et *cicutarium*		*Erodium cicutarium* (L.) L’Hér. ex Aiton	MA072284
77	*Geranium*		*Erodium botrys* (Cav.) Bertol.	MA252575
78	*Geranium purpureum* Vill.		*Geranium* cf. *robertianum* L.	MA071381
79	*Ononis pedunculata* Link		*Ononis broteriana* DC.	MA060978
80	*Ononis reflexa* Link		*Ononis pubescens* L.	MA061642
82	*Genista ericetorum* Link		*Genista* gr. *anglica* L.	MA058777
85	*Genista pendulina* Lam.		*Cytisus striatus* (Hill) Rothm..	MA060415
86	*Quercus australis*	Yes	*Quercus faginea* subsp. *broteroi* (Cout.) A. Camus	MA054711
94	*Statice pungens* Link	Yes	*Armeria pungens* (Link) Hoffmanns. & Link	MA174257
95	*Sium sylvatium* Link		*Physospermum cornubiense* (L.) DC.	MA087694
96	*Eryngium juressi* [*juresianum*]		*Eryngium duriaei* J. Gay ex Boiss.	MA085035
96	*Eryngium juressi* [*juresianum*]		*Eryngium duriaei* J. Gay ex Boiss.	MA085040
99	*Lavatera trimestris*		*Lavatera trimestris* L.	MA076967
101	*Lavatera cretica*		*Lavatera cretica* L.	MA077161
**No letter of reference**				
33	*Lepidium nudicaule*		*Teesdalia coronopifolia* (J.P. Bergeret) Thell.	MA043772
47	_		*Fimbristylis bisumellata* (Forssk.) Bubani	MA234540
62	*Stipa gigantea* Link	Yes	*Stipa gigantea* Link	MA007273
111	*Cynoglossum clandestinum*		*Cynoglossum clandestinum* Desf.	MA095057
114	*Erica carnea*		*Erica cinerea* L.	MA090059
115	*Erica ciliaris*		*Erica ciliaris* Loefl. ex L.	MA089924
116	*Erica arborea*		*Erica arborea* L.	MA090439
117	*Erica*		*Erica umbellata* Loefl. ex L.	MA090206
146	*Statice ferulacea*		*Limonium ferulaceum* (L.) Chaz.	MA151203
_	*Euphorbia pterocarpa*		*Euphorbia* sp.	MA150474
_	*Lycopodium helveticum*		*Selaginella denticulata* (L.) Spring	MA234063
_	*Plantago*		*Plantago lanceolata* L.	MA432267
_	*Scilla*		*Scilla hyacinthoides* L.	MA021620
_	*Valeriana lusitanica* Lag.		*Valeriana tuberosa* L.	MA119545
_	_		*Carex divisa* Hudson	MA237044
_	_		*Echinospartum ibericum* Rivas Mart., Sánchez Mata & Sancho	MA058546
_	_		*Erodium cicutarium* (L.) L´Hér. ex Aiton	MA072199
_	_	Yes	*Orchis italica* Poir. in Lam.	MA237044
_	_		*Plantago albicans* L. ?	MA432316

## Data Availability

The datasets generated in the current research are available from the corresponding author on reasonable request.

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
