# Peer review of "Vascular Plants from the Journey through Portugal (1797–1801) by Hoffmannsegg and Link at the Herbarium of the Real Jardín Botánico of Madrid"

_plants, 2022, doi:10.3390/plants11182438_

Round 1

Reviewer 1 Report

The manuscript "Vascular Plants from the Journey through Portugal (1797–1801)  by Hoffmannsegg and Link at the Herbarium of the Real Jardín  Botánico of Madrid" presents information about four lectotypes (one of which designates a neotype), located in the herbarium of the Real Jardín Botánico of Madrid as a result of analyses of specimen's labels and collectors' correspondence. This is valuable botanical nomenclature data.

Comments:

1 - A map with markings of the locations would be a good illustration of the text and a good background for a graphic abstract.

2 - Please list the names of the four lectotypes in the abstract

3 - Please add a conclusion part highlighting the lectotypes

Author Response

Reviewer 1

Comment 1. Map. Unfortunately, H&L do not provide precise localities that can be represented on a map, so the materials cannot be georeferenced to prepare a map as requested by reviewer 1.

Comment 2. Lectotypes in the abstract. The names of the 4 lectotypes have been added to the abstract.

Comment 3. Conclusion paragraph. We have added a new paragraph about lectotypes as 3.4.

Reviewer 2 Report

The manuscript is well written and very thorough. As such, I do not have any major concerns with it but I do think two points should be clarified: - l. 253: Airochla caudata Link. In Table 1 as well as in the text that follows, it is referred to as Aira caudata Hofmannsegg. Could the authors add a word or two explaining (or at least acknowledging) this discrepancy? - l. 393: similarly, and maybe more importantly, Stipa gigantea Link: is there any explanation as to why Cavanilles wrote Avena gigantea sp. n. on the label? I am not a specialist of this group so I have to trust the authors that this is indeed the same species, but as it is written it reads as if it could be two different species with the same epithet. An explanation would be a welcomed addition here. In addition, those two more minor points: - l. 63 B: i get that this stands for Berlin but the acronym was not introduced before. - l. 72 tree = three? Apart from those very minor points, I do not see any issues that would preclude this manuscript from being published.

Author Response

Reviewer 2

Comment 1. Aira caudata (I. 253). The name Aira caudata Hoffmannsegg is a provisional name on the label, which is not validly published. When Link publishes this name it decides to do so under the Airochloa genus. We have added the correct name in table 1, following Hoffmannsegg's name, to avoid any ambiguity.

Comment 2. Stipa (i. 393). We have added an explanation phrase about this item: <<This specimen is connected to Link species through the epithet "gigantea" and fits very well with the current concept of S. gigantea>>.

Comment 3. Corrections. I. 63 and I. 72 mistakes have been corrected.

Reviewer 3 Report

It is an interesting study of historical collections that brings a notable systematic contribution. Both the introduction and the study methods are appropriate. The description is very accurate and the photos are particularly interesting. The bibliography is complete.

Author Response

No suggestions indicated by the reviewer 3

Thanks

Reviewer 4 Report

No suggestions.

Author Response

No suggestions indicated by the reviewer 4

Thanks